



# Climatological distribution of ocean acidification indicators along the North American ocean margins

Li-Qing Jiang[1,2], Tim P. Boyer[2], Christopher R. Paver[2], Hyelim Yoo[1,2], James R. Reagan[2], Simone R. Alin[3], Leticia Barbero[4,5], Brendan R. Carter[3,6], Richard A Feely[3], and Rik Wanninkhof[4].

[1]Cooperative Institute for Satellite Earth System Studies, Earth System Science Interdisciplinary Center, University of Maryland, College Park, Maryland 20740, United States
[2]NOAA/NESDIS National Centers for Environmental Information, Silver Spring, Maryland 20910, United States
[3]NOAA/OAR Pacific Marine Environmental Laboratory, Seattle, Washington 98115, United States
[4]NOAA/OAR Atlantic Oceanographic and Meteorological Laboratory, Miami, Florida 33149, United States
[5]Cooperative Institute for Marine and Atmospheric Studies, Rosenstiel School of Marine and Atmospheric Science, University of Miami, 4600 Rickenbacker Causeway, Miami, Florida 33149, United States
[6]Cooperative Institute for Climate, Ocean, and Ecosystem Studies, University of Washington, Seattle, WA 98105, United States

*Correspondence to*: Li-Qing Jiang (Liqing.Jiang@noaa.gov)

**Abstract.** Climatologies, which depict mean fields of oceanographic variables on a regular geographic grid, and atlases, which provide graphical depictions of specific areas, play pivotal roles in comprehending the societal vulnerabilities linked to ocean acidification (OA). This significance is particularly pronounced in coastal regions where most economic activities related to commercial and recreational fisheries as well as aquaculture industries occur. In this paper, we unveil a comprehensive data product featuring coastal climatologies and atlases for ten OA indicators, including fugacity of carbon dioxide, pH on the total scale, total hydrogen ion content, free hydrogen ion content, carbonate ion content, aragonite saturation state, calcite saturation state, Revelle Factor, total dissolved inorganic carbon content, and total alkalinity content. These indicators are provided on 1°×1° degree spatial grids at 14 standardized depth levels, ranging from the surface to a depth of 500 meters, along the North American ocean margins — defined as the region between the coastline and a distance of 200 nautical miles (∼370 km) offshore. The climatologies and atlases were developed using the World Ocean Atlas (WOA) gridding methods of the NOAA National Centers for Environmental Information (NCEI), based on the recently released Coastal Ocean Data Analysis Product in North America (CODAP-NA), along with the 2021 update to the Global Ocean Data Analysis Project version 2 (GLODAPv2.2021) data product. The relevant variables were adjusted to the index year of 2010. The data product is available in NetCDF (DOI: 10.25921/g8pb-zy76) at the NOAA Ocean Carbon and Acidification Data System: https://www.ncei.noaa.gov/data/oceans/ncei/ocads/metadata/0270962.html. It is recommended to use the objectively analyzed mean fields (with "_an" suffix) for each variable. The atlases can be accessed at: https://www.ncei.noaa.gov/access/ocean-carbon-acidification-data-system/synthesis/nacoastal.html.



## 1 Introduction

The chemistry of the mildly alkaline ocean has been changing as a result of absorbing ~25% of the carbon dioxide ($CO_2$)
released by human activities (Gruber et al., 2019; Jiang et al., 2019; DeVries, 2022; Friedlingstein et al., 2023; Ma et al.,
2023; Feely et al., 2023; Jiang et al., 2023; Richardson et al., 2023). This process, which is causing an increase in ocean
acidity and a decrease in the substance content of carbonate ion (a building block for shells and skeletal structure of many
marine organisms), is commonly referred to as ocean acidification (OA) (Caldeira and Wickett, 2003; Feely et al., 2004; Orr
et al., 2005; Doney et al., 2009; Gattuso and Hansson, 2011). Studies have shown that OA can negatively affect marine
organisms that form shells and skeletons using calcium carbonate ($CaCO_3$, with aragonite and calcite as the dominant
mineral forms), and has the potential to significantly impact shellfish fisheries and aquaculture (Cooley and Doney, 2009;
Andersson and Gledhill, 2013; Gattuso et al., 2015; Albright et al., 2016; Connell et al., 2018; Kawahata et al., 2019; Doney
et al., 2020).

Despite occupying just 7% of the ocean surface area, coastal seas (< 200 meters deep near land) are among the most
productive parts of the global ocean, accounting for 90% of global fisheries yield (Tickler et al. 2018). The coastal ocean
also contains some of the richest biodiversity (80% of known species of marine fish, Cicin-Sain et al., 2002), and provides
important ecosystem goods and services to billions of people, in the form of food security, fishery and aquaculture
industries, and recreational activities, worth more than $27.7 trillion U.S. dollars annually, a number larger than the annual
U.S. gross domestic product (de Groot et al., 2012, Costanza et al., 2014; Kubiszewski et al., 2017). Thus, understanding the
status of OA in the coastal ocean is critical to guide society's OA mitigation and adaptation efforts.

In a recent study, two decades of discrete measurements of inorganic carbon system parameters, oxygen, and nutrient
chemistry data from the North American ocean margins were compiled, quality controlled (QCed), and synthesized to
generate a data product called the Coastal Ocean Data Analysis Product in North America (CODAP-NA) (Jiang et al., 2021).
CODAP-NA makes the QCed cruise data available in various uniform formats, e.g., CSV, Excel, NetCDF and MATLAB,
facilitating future OA research in the North American ocean margins. However, it does not provide their values on
standardized spatial grids and depth levels.

In this companion paper, we describe the climatologies (mean fields of oceanographic variables on a regular geographic
grid) and atlases (maps of these properties for the area of interest) at 14 standard depth levels from the surface to 500 meters
for fugacity of carbon dioxide ($f$CO$_2$), pH on the total scale (pH$_T$), total hydrogen ion content ($[H^+]_{total}$), free hydrogen ion
content ($[H^+]_{free}$), carbonate ion content ($[CO_3^{2-}]$), aragonite saturation state ($\Omega_{arag}$), calcite saturation state ($\Omega_{cal}$), Revelle
Factor (RF), total dissolved inorganic carbon content (DIC), and total alkalinity content (TA). The generated climatologies
are available in NetCDF, and the atlases are available in still images (jpeg). The produced climatologies and atlases provide





a baseline for current OA conditions for use in assessing future changes along the North American ocean margins, enabling the identification of more vulnerable vs. potentially resilient regions. Additionally, the climatologies will facilitate regional model validation, allow users to plot the distribution of an OA indicator on horizontal or vertical sections in different regions easily, and make it possible to extract values at specific longitude, latitude, and depth combinations. The atlas visualizations

of these parameters will inform coastal enterprises, marine resource decision makers, and the general public about the current status of OA in each region, so as to provide actionable information for the coastal mitigation and adaptation efforts.

## 2 Technical Approach and Methodology

Climatologies and atlases for a total of ten OA indicators: $f\mathrm{CO_2}$, $\mathrm{pH_T}$, $[\mathrm{H^+}]_{total}$, $[\mathrm{H^+}]_{free}$, $[\mathrm{CO_3^{2-}}]$, $\Omega_{arag}$, $\Omega_{cal}$, RF, DIC, and TA, along with temperature and salinity, were created in the ocean margins of North America (Table 1). Input data came

primarily from the QCed cruise data from CODAP-NA, which contains discrete measurements throughout the water column of the coastal region (Jiang et al., 2021), and the 2021 update to the Global Ocean Data Analysis Project (GLODAPv2.2021), which includes discrete measurements from the open ocean (Lauvset et al., 2021), both spanning from the ocean surface to the seafloor. The relevant inorganic carbon system variables were indexed to the year of 2010 based on the algorithms developed by Carter et al. (2021). Specifically, "ESPER_Mixed" was used to estimate the delta dissolved inorganic carbon

(DIC) differences between the sampling year and 2010. For more details of this temporal adjustment, refer to Jiang et al. (2023). The carbonate system calculations were conducted using a Julia version (CO2System.jl v2.0.5, Humphreys et al., 2022) of the CO2SYS program (Lewis and Wallace, 1998), with the dissociation constants for carbonic acid of Lueker et al. (2000), bisulfate ($\mathrm{HSO_4^-}$) of Dickson (1990), hydrofluoric acid (HF) of Perez and Fraga (1987), and with the total borate equations of Lee et al. (2010) as recommended by Jiang et al. (2022). When more than two carbonate chemistry variable

measurements were available for a profile, the variables used for carbonate chemistry calculations were chosen based on the preference DIC>TA>pH. This preference order corresponds to the frequency of availability of these measurement types in the underlying data products and is therefore chosen on the basis of maximizing the consistency of the underlying calculations. It has been noted that the stated uncertainty in seawater pH measurements is often low enough to result in reduced uncertainties in calculations using this variable compared to the other variables, but we nevertheless use this

ordering to ensure our underlying calculations are maximally consistent, and because of lingering issues with accuracy of pH at depth (Carter et al., 2023).

$[\mathrm{H^+}]_{total}$ and $[\mathrm{H^+}]_{free}$ (unit: $10^{-9}$ mol kg$^{-1}$) were directly calculated from a Julia version of the CO2SYS (CO2System.jl). Because pH is on a logarithmic scale, it was not gridded directly. Instead, its corresponding $[\mathrm{H^+}]_{total}$ was gridded, the gridded

pH values were then calculated using the definition of pH (Equation 1).

$$\mathrm{pH_T} = -\log_{10} [\mathrm{H^+}]_{total} \qquad\qquad (1)$$



Note that in Equation (1), the unit for $[H^+]_{total}$ is moles per kilogram. Saturation state of carbonate minerals (unitless) is defined as:

$$\Omega = \frac{[Ca^{2+}] \times [CO_3^{2-}]}{K'_{sp}} \tag{2}$$

where $\Omega$ is the saturation state ($\Omega > 1$ favors precipitation and $\Omega < 1$ favors dissolution), $[Ca^{2+}]$ and $[CO_3^{2-}]$ are the calcium and carbonate ion contents, respectively (units: $\mu mol\ kg^{-1}$). $K'_{sp}$ is the apparent solubility product of the calcium carbonate minerals (e.g., aragonite or calcite). $[Ca^{2+}]$ in seawater was assumed to be conservative with salinity according to *Millero* [1995]. The Revelle Factor (RF) quantifies the buffer capacity for the seawater carbonate system, defined by Revelle and Suess (1957) as the ratio of the fractional change in $f\text{CO}_2$ to the fractional change in DIC, with TA constant. A
higher RF indicates a smaller DIC change (or the amount of carbon the ocean absorbs) for a specific change in sea surface $f\text{CO}_2$. A higher RF also suggests a given addition of DIC will have a larger impact on $f\text{CO}_2$, implying the seawater is less well-buffered (Broeker et al., 1979).

**Table 1.** Variables included in this data product. All of them are reported at in-situ temperature and pressure. The leftmost
column shows the abbreviations used in the NetCDF files.

| Abbreviation | Full variable name | Unit |
|:---:|:---:|:---:|
| fCO2 | Fugacity of carbon dioxide | $\mu atm$ |
| pHT | pH on total scale | unitless |
| Htotal | Total hydrogen ion content | $nmol\ kg^{-1}$ |
| Hfree | Free hydrogen ion content | $nmol\ kg^{-1}$ |
| CO3 | Carbonate ion content | $\mu mol\ kg^{-1}$ |
| OmegaA | Aragonite saturation state | unitless |
| OmegaC | Calcite saturation state | unitless |
| RF | Revelle Factor | unitless |
| DIC | Total dissolved inorganic carbon content | $\mu mol\ kg^{-1}$ |
| TA | Total alkalinity content | $\mu mol\ kg^{-1}$ |
| T | Water temperature | °C |
| S | Salinity (Practical Salinity Scale of 1978) | unitless |



For all OA indicators, the data were vertically interpolated onto 14 standardized depth levels: 0, 10, 20, 30, 50, 75, 100, 125, 150, 200, 250, 300, 400, 500 meters, before they were horizontally gridded. These standardized depth levels are the same as those used in GLODAPv2 gridded data product (Lauvset et al., 2016), as originally chosen by Levitus and Boyer (1994).

Vertical interpolation was performed using the 4-point Reiniger-Ross interpolation approach when data permitted, otherwise a 3-point Lagrangian interpolation was used, with linear interpolation serving as a last resort (Reiniger and Ross, 1968). The algorithm used to calculate a complete gridded field at specified standard depths in the ocean is based on the objective analysis technique, initially developed for atmospheric variables by Barnes (1964), and adapted for oceanographic applications (non-continuous globally due to land and ocean bottom) by Levitus (1982). The technique adjusts a first-guess

(best estimate of value at each grid/depth) field based on the weighted difference between the first guess and the value of the mean of all observations at each standard depth (vertically interpolated from observations if necessary) for each grid-box within a specified radius of influence (distance) around a specific grid box. The first-guess fields for each of these variables at all depth levels were calculated using the Empirical Seawater Property Estimation Routines (ESPERs) algorithms (Carter et al., 2021) based on the WOA 2018 data for salinity, temperature, and dissolved oxygen (Zweng et al., 2018; Locarnini et

al., 2018; Garcia et al., 2018). The radius of influence is set based on physical ocean forcing such as the Rossby radius. The technique was chosen because of its emphasis on preserving gradients associated with ocean physics at the chosen grid resolution and because of the adaptability of the method for data-sparse regions without sufficient information for robust covariances (Reagan et al. 2023). The atlases of this product were produced using the Generic Mapping Tools (GMT) (Wessel et al., 2019). See Figure 1 as an example.

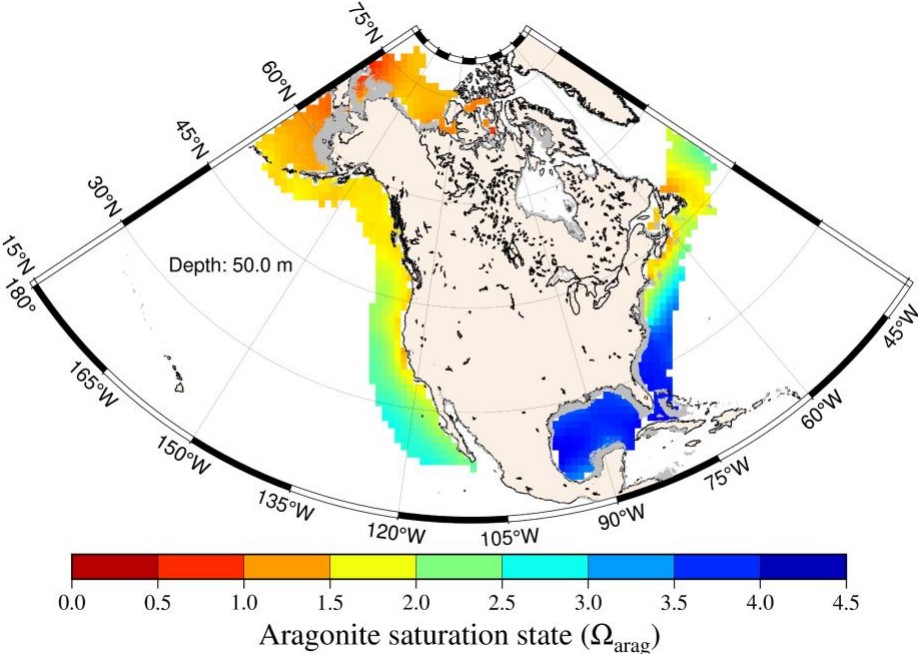


**Figure 1.** Aragonite saturation state at 50 meters depth on the North American ocean margins.



The methodology employed in generating these coastal carbon climatologies aligns with the methods utilized for the production of the World Ocean Atlas 2023 (Reagan et al., 2023). In addition to the objectively analyzed means (recommended), other gridding parameters, e.g., statistical mean, number of observations, standard deviation, standard error, etc. were also generated and made available as part of this product. See Table 2 for their definitions.

**Table 2.** Gridding parameters for each variable within this data product. The leftmost column shows the abbreviations used in the NetCDF files.

| Abbre-viation | Full name | Descriptions |
|---|---|---|
| _an | Objectively analyzed mean | The mean fields of an oceanographic variable at standard depth levels of the global ocean (recommended). |
| _mn | Statistical mean | The average of all depth-interpolated data values that pass quality control checks at each standard depth level for each variable within the one-degree square, which contain at least one measurement for the given oceanographic variable. |
| _dd | Number of observations | The count of quality-controlled observations for each variable within the one-degree square at each standard depth level. |
| _sd | Standard deviations | The spread of each variable within the one-degree square at each standard depth level that passes quality control checks. |
| _se | Standard errors | The errors as defined in Levitus et al. (2012) for each variable in the one-degree square at each standard depth level that passes quality control checks. |
| _gp | Number of grid_squares | The number of one-degree squares within the smallest radius of influence around each one-degree square that contains a statistical mean value. |

## 3 Data availability

The produced climatologies (gridded data at standard depth levels) are available in NetCDF through NCEI's archive [NCEI Accession Number: 0270962, DOI: 10.25921/g8pb-zy76]:

https://www.ncei.noaa.gov/data/oceans/ncei/ocads/metadata/0270962.html.

Their corresponding atlases (plotted color maps based on the gridded data) of these variables at 14 standardized depth levels from surface to 500 meters are accessible through a web interface: https://www.ncei.noaa.gov/access/ocean-carbon-acidification-data-system/synthesis/nacoastal.html.



## 4. Summary

In this study, we utilized the World Ocean Atlas (WOA) gridding methodologies from NOAA's National Centers for Environmental Information (NCEI) to generate a data product featuring climatologies and atlases on $1° \times 1°$ grids for North American ocean margins. This product showcases 10 OA indicators across 14 standardized depth levels, ranging from the surface to 500 meters. The OA variables comprise: fugacity of carbon dioxide ($f\text{CO}_2$), pH on the total scale ($\text{pH}_{total}$), total hydrogen ion content, free hydrogen ion content, carbonate ion content, aragonite saturation state, calcite saturation state, Revelle Factor, total dissolved inorganic carbon content, total alkalinity content, as well as temperature and salinity. The depth levels are set at: 0, 10, 20, 30, 50, 75, 100, 125, 150, 200, 250, 300, 400, and 500 meters. The basis for this work is the recently released CODAP-NA and GLODAPv2.2021, which encompasses 3391 oceanographic profiles from 61 research cruises, including 18,341 DIC measurements, and 18,351 TA measurements. All variables were adjusted to the year of 2010 before the gridding process.

**Author contribution:**

All authors contributed to the writing of the paper. L-QJ coordinated with the overall effort, performed the carbon system calculations, created the plots, and prepared the initial draft of the paper. TPB programmed the World Ocean Atlas tools and mentored the team on gridded product creation. CRP developed the climatologies and managed quality control for numerous parameters, including pH on the Total Scale, total and free hydrogen ions, carbonate ion, aragonite and calcite saturation states, and both total dissolved inorganic carbon and alkalinity. In a similar capacity, HY processed fugacity of carbon dioxide and the Revelle Factor. JRR and HY worked together to extract the climatologies for temperature and salinity out of the existing WOA products. JRR offered essential programming assistance for the development of these climatologies. SRA, LB, BRC, RAF, and RW (listed alphabetically by surname) contributed data, determined the calculation and gridding approach, and provided guidance to L-QJ throughout the process on this overall effort.

**Competing interest**

The authors declare that they have no conflict of interest.

**Acknowledgements**

We thank Alex Kozyr (University of Maryland, partially funded by the National Oceanic and Atmospheric Administration (NOAA) Global Ocean Monitoring and Observing Program) for helping archive the final data product. We are grateful to Liem Nguyen (University of Maryland) for helping develop the visualization web interface. This is NOAA Pacific Marine





Environmental Laboratory (NOAA/PMEL) contribution number 5605 and the University of Washington Cooperative
Institute for Climate, Ocean, & Ecosystem Studies (UW/CICOES) Contribution No. 2024-1343.

**Financial support**

Funding for L-QJ is from NOAA Ocean Acidification Program (OAP, https://ror.org/02bfn4816) and NOAA National
Centers for Environmental Information (NCEI) through a NOAA Cooperative Institute for Satellite Earth System Studies
(CISESS) grant (NA19NES4320002) at the Earth System Science Interdisciplinary Center (ESSIC), University of Maryland.
SRA and RAF thank NOAA Pacific Marine Environmental Laboratory for salary support. BRC thanks the Carbon Data
Management and Synthesis Grant (Fund Ref: 100007298) from the NOAA's Global Ocean Monitoring and Observation
division. This publication was partially funded by the Cooperative Institute for Climate, Ocean, & Ecosystem Studies
(CICOES), University of Washington, under NOAA Cooperative Agreement NA20OAR4320271.



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
