# Peer review of "Climatological distribution of ocean acidification variables along the North American ocean margins"

_Earth System Science Data, 2024_

## Author Response (AR1)

**Responses to Reviewers' Comments**

**RC1**: 'Comment on essd-2024-59', Abed El Rahman Hassoun, 16 Apr 2024

Jiang et al. have presented the climatological distribution of ocean acidification indicators along the North American ocean margins. The data are very interesting and useful for the community. Therefore, I would recommend the publication of this ms. after addressing a couple of comments.

**Response:** Many thanks for your time in reviewing this manuscript and for the positive comments.

**Major comments:**

P1, L45: You are mentioning that "coastal seas (< 200 meters deep near land)". According to which definition/source are you relying?

**Response:** In this paper, we defined the ocean margins as "between the coastline and a distance of 200 nautical miles (~370 km) offshore".

This definition is based on:
  (a) The United Nations Convention on the Law of the Sea:
      https://en.wikipedia.org/wiki/United_Nations_Convention_on_the_Law_of_the_Sea, and
  (b) The U.S. Presidential Proclamation No. 5030 of March 10, 1983:
      https://www.gc.noaa.gov/documents/031483-proc_5030_48fr10605.pdf.

There is only one figure in the manuscript, visualizing aragonite saturation state at 50 meters depth. I would suggest to add more figures showing the patterns of other parameters/indicators as well.

**Response:** Another figure showing the distribution of Revelle Factor at 100 meters has been added. See Figure 2.

**Minor comments:**

It's more conventional to refer to the following terms used in this ms., such as "inorganic carbon system variables" "carbonate chemistry variable" "carbonate chemistry", as the carbonate system (parameters). Please change that throughout the ms.

**Response:** All instances of "inorganic carbon system" and "carbonate chemistry", have now been changed to "carbonate system".

Please unify the terminology used for the carbonate ions. Sometimes you use carbonate minerals or carbonate ion.

**Response:** We have thoroughly reviewed the manuscript. Although both carbonate minerals and carbonate ions were mentioned, they were used to denote distinct entities: carbonate minerals for CaCO3, whereas carbonate ions for CO32-.

**Abstract:**

P1, L17-18: Economic activities are not restricted to fisheries and aquaculture. Thus, I suggest to change the sentence as follows: "This significance is particularly pronounced in coastal regions where most economic activities, such as commercial and recreational fisheries and aquaculture industries, occur."

**Response:** The change has been made accordingly. See the new sentence on Line 17.

**Introduction:**

P2, L52-53: I suggest editing the sentence as follows: "...measurements of the carbonate system parameters, oxygen, and nutrients' data...".

**Response:** The change has been made accordingly. See the new sentence on Line 53.

P3, L70: Please replace "marine resource decision makers" by "decision-makers and stakeholders".

**Response:** Changes have been made accordingly. See Line 70.

**Methodology:**

Please check again the uniformity of citations in the ms. For example, in P5, L128, it should be (Reagan et al., 2023).

**Response:** We have double checked all the citations to ensure they are in the same format.

===================================

**RC2**: 'Comment on essd-2024-59', Anonymous Referee #2, 17 Apr 2024
**Summary**

Jiang et al. provide a suite of standardized climatologies and atlases of the coastal North American margin for the year 2010, created from the CODAP-NA and GLODAPv2.2021 data products. These climatologies and atlases are well-described and easy to access and use. I would recommend publication of this paper after addressing the following comments.

**Response:** Many thanks for your time in reviewing this manuscript and for the positive comments.

**Major comments**

I have no major concerns with this paper and, in general, appreciate the brevity of it. However, there are two instances where I think the authors should expand the manuscript to add more information and discussion for the audience.

First, I recommend the authors add a few additional details about the spatial interpolation method to create a gridded field (circa lines 115-125). In particular, I was left with the following questions: How exactly were the "best-guess" estimates adjusted by the weighted means of the surrounding cells? Were the surrounding cells weighted by distance or some other factor? Was the radius of influence always the Rosby radius for the region? If not, what alternative was used and what criteria did the authors apply to determine the most appropriate radius?

**Response:** Significant details have been added. See the 3 new paragraphs on Page 6.

Secondly, I think this paper would be strengthened with the addition of a few sentences or a paragraph discussing how the authors anticipate these static, annual climatologies indexed to 2010 to relate to current and sub-annual conditions. Are the spatial patterns of these climatologies assumed to be stable through time? Are these climatologies relatable to winter conditions, given that most cruises in CODAP-NA took place in the summer months? These considerations are particularly relevant with respect to the argument in line 70 that these climatologies and atlases will provide insight into the "current status of OA in each region."

**Response:** Several sentences have been added. See Page 6.

"While these climatologies offer extensive spatial and vertical coverage, they provide

only a static view of OA conditions referenced to 2010. Insufficient observational data precluded the delineation of seasonal or sub-annual variations. Furthermore, the data used for generating these climatologies were primarily collected during the summer and fall seasons."

**Minor comments**

- Line 47: Remove the comma after "biodiversity".

**Response:** Done. See Line 47 for the updated sentence.

- Line 48: Remove the comma after "people".

**Response:** Done. See Line 48 for the updated sentence.

- Line 49: I recommend breaking this into two sentences, ending the first after "activities."

**Response:** Done. See Lines 46 - 51 for the updated sentences.

==========================

**CC1**: 'Comment on essd-2024-59', Su Kyong Yun, 22 Apr 2024

This data description paper introduces ocean acidification (OA) indicators, comprising climatologies and atlases, which serve as valuable resources for various stakeholders and researchers aiming to enhance their understanding of OA in North America. The indicators are effectively presented, with the table providing clear information on units and the description of the calculation method. The atlas clearly describes the regions covered by the data with the scale.

**Response:** Many thanks for your time in reviewing this manuscript and for the positive comments.

A minor suggestion would be to include details in the variable table indicating whether the variables are measured or calculated, along with the methods used for their calculation. This addition would enhance users' understanding of the data.

**Response:** We have added a new column to Table 1 to address this excellent comment.

Additionally, specifying the datum used for standardized depth levels would add clarity.

**Response:** Not sure if we fully understand this comment, but for each variable, it was used to generate their values in these standardized levels according to the sentence below:

"Vertical interpolation was performed using the 4-point Reiniger-Ross interpolation approach when data permitted, otherwise a 3-point Lagrangian interpolation was used, with linear interpolation serving as a last resort (Reiniger and Ross, 1968)."

Or if this is about the standardized depth levels themselves, we adopted them from the World Ocean Atlas.

Overall, the paper demonstrates excellent organization and is written with meticulous attention to detail.

**Citation**: https://doi.org/10.5194/essd-2024-59-CC1